# MED3D-JADE: 3D JOINT ATTENTIVE DIFFUSION ENGINE FOR VOLUMETRIC MEDICAL CT AND MASK CO-GENERATION

## ABSTRACT

Data scarcity is a critical bottleneck for training robust 3D medical image segmentation models. Current generative approaches for paired data synthesis are often limited to conditional generation (e.g., mask-to-image), which cannot produce novel anatomical structures and thus fail to address the lack of structural diversity in training data. To overcome this, we introduce **Med3D-JADE**, the first diffusion-based framework, to our knowledge, that learns the true joint distribution $p(\text{image, mask})$ to simultaneously generate entirely new 3D CT volumes and their corresponding segmentation masks. Our method adapts a pre-trained 3D medical generation foundation model (MAISI) into a dual-branch latent diffusion architecture. We preserve the foundation model's high-fidelity image synthesis by freezing its original branch and training a new, parallel branch for segmentation. Our proposed **Volumetric Joint Attention (VJA)** modules enforce coherence between the modalities , while the reuse of the model's powerful Volumetric Compression Network (VCN) facilitates efficient, high-resolution generation for both domains without needing to train a new encoder from scratch. We rigorously validate our approach by using the generated pairs for data augmentation across four datasets, including public benchmarks like SegTHOR and challenging MSD tumor datasets. Augmenting with our synthetic data leads to significant performance gains for diverse segmentation models like nnU-Net, SwinUNETR, and SegResNet, establishing Med3D-JADE as a generalizable and practical solution for overcoming 3D data scarcity in medical imaging.

## 1 INTRODUCTION

The performance of deep learning models for 3D medical image segmentation is critically limited by data scarcity, a problem exacerbated by the high cost of expert volumetric annotations . While generative models offer a path for data augmentation, the dominant paradigm of **conditional synthesis**—learning to generate an image from a given mask by modeling $p(\text{image} — \text{mask})$—has a fundamental flaw . Such models can only produce new textural variations for a fixed set of anatomical structures from the training data; they cannot generate novel shapes or configurations, failing to address the core need for greater **structural diversity** .

We argue that a more powerful approach is to learn the true **joint distribution**, $p(\text{image, mask})$, enabling the simultaneous generation of entirely new 3D CT volumes and their corresponding masks . This unlocks the ability to create true structural novelty, which is impossible with conditional methods . However, extending joint generation to 3D medical volumes is a formidable challenge . The massive increase in computational complexity, memory requirements, and the need to maintain spatial coherence across hundreds of slices has pushed prior work towards simpler conditional or less efficient two-stage paradigms . A true, single-stage joint generation framework for 3D medical data has remained an open problem .

To address this challenge, we introduce **Med3D-JADE**, a novel framework that learns the joint distribution for 3D CT and mask co-generation. We propose a dual-branch latent diffusion architecture that efficiently adapts a powerful, pre-trained image generator by freezing its weights to preserve quality while training a parallel branch for segmentation . Cross-modal coherence is enforced by

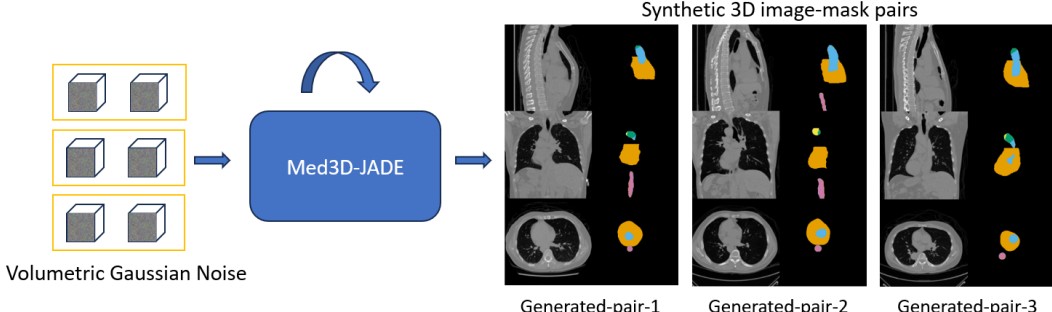

Figure 1: **Overview of Med3D-JADE: Joint Generation of Diverse and Coherent 3D Medical Image-Mask Pairs.** Med3D-JADE takes volumetric Gaussian noise as input and, through a novel Volumetric Joint Attention (VJA) mechanism, simultaneously synthesizes both a realistic 3D CT image and its corresponding, anatomically aligned segmentation mask. This process ensures semantic coherence between the generated image and its structure. The output consists of diverse, high-fidelity image-mask pairs that significantly enrich medical training datasets. Here, three distinct volumetric noise inputs yield three unique synthetic image-mask pairs, demonstrating the model's ability to generate varied and clinically plausible anatomical configurations. Each row showcases different axial, sagittal, and coronal slices from a single generated 3D pair.

our core contribution, a **Volumetric Joint Attention (VJA)** mechanism . Our main contributions are:

- To our knowledge, we present the **first framework** for the simultaneous, single-stage joint generation of 3D medical volumes and their corresponding segmentation masks from random noise .

- We propose a novel **dual-branch architecture with a Volumetric Joint Attention (VJA)** mechanism to ensure cross-modal coherence while efficiently adapting a pre-trained foundation model .

- We demonstrate that the superior **structural diversity** from our joint model directly translates to significant performance gains for downstream segmentation tasks, validating our approach as a practical solution for 3D data scarcity .

## 2 RELATED WORK

### 2.1 CONDITIONAL MEDICAL IMAGE SYNTHESIS

The conditional generation paradigm was pioneered by GANs for tasks like image-to-image translation (e.g., Pix2Pix Isola et al. (2017), Pix2PixHD Wang et al. (2018)) and layout-based synthesis (e.g., Layout2Image Zhao et al. (2020), Obj-GAN Li et al. (2019)). This approach was extended to 3D medical imaging with models such as Vox2Vox Cirillo et al. (2021) and HA-GAN Sun et al. (2022), but often faced training instability. This led to a shift towards diffusion models Ho et al. (2020), which offer higher fidelity and stability. In 3D medical imaging, this has resulted in a variety of conditional diffusion models focused on efficiency (e.g., 3D MedDiffusion Wang et al. (2024), Make-A-Volume Zhu et al. (2023)) and diverse conditioning strategies (e.g., Med-DDPM Dorjsembe et al. (2024), Diff-Boost Zhang et al. (2024c), GEM-3D Zhu et al. (2024)). Among the most powerful is MAISI Guo et al. (2025), a large-scale framework adapting a ControlNet-like architecture Zhang et al. (2023) for detailed, mask-based conditioning. While some models like Medical Diffusion Khader et al. (2022) can generate volumes unconditionally, the majority of high-fidelity methods model the conditional distribution $p$(image — condition), fundamentally limiting them to textural variations rather than novel structures.

## 2.2 Joint Image and Annotation Synthesis

To overcome this limitation, research has shifted towards joint synthesis, which learns the joint distribution $p(\text{image, annotation})$ to generate entirely new data pairs. In general computer vision, two main strategies have emerged. One approach derives annotations by training lightweight decoders on the features of a frozen image generator, as demonstrated by DiffuMask Wu et al. (2023b), DiffusionEngine Zhang et al. (2024b), and DatasetDM Wu et al. (2023a). A second strategy uses specialized architectures for joint modeling. This includes concatenation-based models for 2D data like SatSynth Toker et al. (2024), SimGen Bhat et al. (2025), and CoSimGen Bose et al. (2025). More relevant to our work is the use of coupled dual-branch networks that exchange information, as seen in works like MM-Diffusion Ruan et al. (2023), IDOL Zhai et al. (2024), and JointNet Zhang et al. (2024a). Our framework adapts this dual-branch paradigm to the 3D medical domain.

## 2.3 The Gap in 3D Medical Joint Synthesis

Despite these advancements, joint synthesis for 3D volumetric medical data remains underexplored. The most relevant prior work, MedGen3D Han et al. (2023), uses a two-stage sequential process: it first generates a mask, and then uses that mask to condition a separate image generator. This two-stage approach is inefficient, risks error propagation, and does not model the true joint distribution in a unified manner. This leaves a critical gap for a framework capable of efficient, single-stage, simultaneous co-generation of 3D medical volumes and their masks. Med3D-JADE is the first framework designed to fill this gap.

## 3 Methodology

We introduce **Med3D-JADE**, a novel framework designed to learn the joint probability distribution $p(\text{CT}, \text{Mask})$ for 3D medical volumes. By directly modeling this distribution, our approach is the first to enable the simultaneous, single-stage generation of entirely new volumetric CTs and their corresponding 3D masks, allowing for powerful data augmentation with novel structural diversity.

### 3.1 Preliminary: Foundation Models

Our work builds upon two key concepts: Latent Diffusion Models and a pre-trained 3D medical synthesis framework.

**Latent Diffusion Models (LDMs)**    Our work builds upon Latent Diffusion Models (LDMs) Rombach et al. (2022), which generate high-fidelity data by reversing a noising process in a compressed latent space Ho et al. (2020). An LDM framework typically uses a powerful autoencoder to learn this latent space, where a U-Net is then trained to iteratively denoise a random vector to produce a new sample.

**The MAISI 3D Medical Foundation Model**    Specifically, we build upon MAISI Guo et al. (2025), a state-of-the-art generative foundation model for high-resolution 3D medical image synthesis . MAISI consists of two core components: a Volume Compression Network (VCN) for efficient latent space representation, and a Latent Diffusion Model (LDM) for generating realistic anatomy . These components were trained on massive and diverse medical datasets; the VCN was trained on 39,206 3D CT and 18,827 3D MRI volumes, while the LDM was trained on the latents of 10,277 CT volumes from 24 distinct datasets . Med3D-JADE leverages these powerful, publicly available pre-trained components as its architectural foundation .

### 3.2 Problem Formulation

Let $\mathcal{D} = \{(\mathbf{x}_i, \mathbf{y}_i)\}_{i=1}^{N}$ be a 3D medical dataset, where $\mathbf{x}_i$ is a CT volume and $\mathbf{y}_i$ is its corresponding segmentation mask. Conventional conditional methods learn $p(\mathbf{x}|\mathbf{y})$, a formulation that is fundamentally constrained as it can only generate new textural variations for the finite set of anatomical structures present in $\mathcal{D}$. Our objective is to train a generative model, $\mathcal{G}_\theta$, that approximates the true joint distribution $p(\mathbf{x}, \mathbf{y})$. By sampling from this model, we can generate entirely

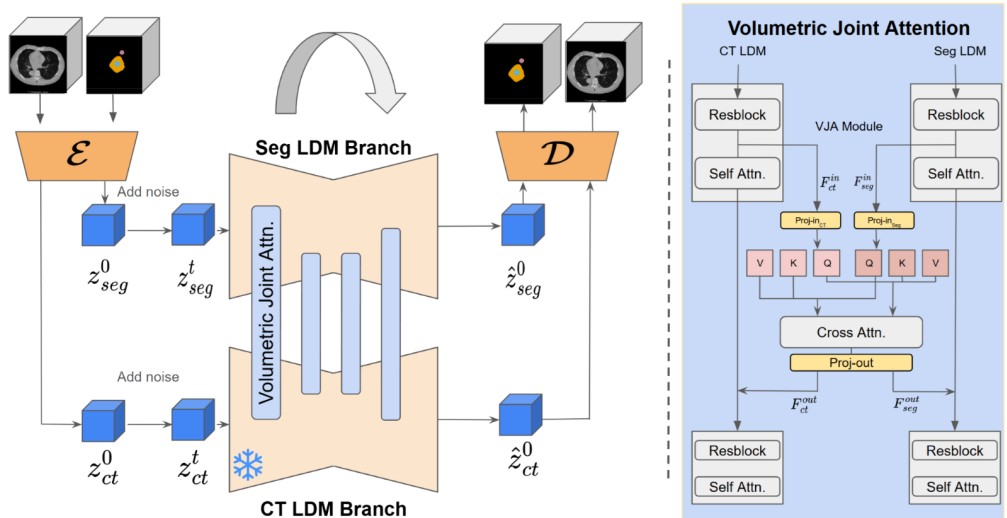

Figure 2: The proposed Med3D-JADE architecture. Our model leverages a pre-trained encoder/decoder (VCN) and a frozen LDM branch (CT LDM) from the MAISI foundation model. A parallel segmentation branch (Seg LDM) is trained, with coherence enforced by our novel Volumetric Joint Attention (VJA) modules that facilitate a bidirectional exchange of information.

new pairs $(\mathbf{x}', \mathbf{y}')$ that exhibit novel structural diversity, creating a richer augmented dataset to train more robust downstream models.

## 3.3 MED3D-JADE: A DUAL-BRANCH FRAMEWORK FOR JOINT GENERATION

A naive approach to joint generation might concatenate the latents of the image and mask and process them with a single LDM. However, this is ill-suited for adapting powerful foundation models like MAISI, as modifying input layers would destroy the learned weights. Inspired by 2D pair-generation work Zhang et al. (2024a), we designed Med3D-JADE as a **dual-branch 3D Latent Diffusion Model** (Figure 2).

Our framework consists of two parallel branches—one for the CT, one for the mask—both initialized with the pre-trained MAISI LDM weights. We employ a parameter-efficient fine-tuning strategy: the entire **CT branch remains frozen** to preserve its high-fidelity synthesis capabilities, while updates are exclusively applied to the **segmentation branch** and our proposed **Volumetric Joint Attention (VJA)** modules. For efficiency, we reuse MAISI's single pre-trained **Volume Compression Network (VCN)** for both modalities. To enable this, we normalize the discrete mask labels into a continuous grayscale representation in the range $[0, 1]$, allowing the VCN's encoder $E$ to map both CT and normalized mask to the latent space: $\mathbf{z}_{ct} = E(\mathbf{x}_{ct})$ and $\mathbf{z}_{seg} = E(\mathbf{y}_{norm})$. After denoising, the continuous mask output is mapped back to the nearest discrete label.

## 3.4 VOLUMETRIC JOINT ATTENTION (VJA)

The VJA module is our core innovation for enforcing cross-modal coherence. A VJA module is inserted into the 3D U-Net of both branches at each level where a self-attention block was originally present in MAISI. The module performs a bidirectional information exchange. Given feature maps $\mathbf{F}_{ct}$ and $\mathbf{F}_{seg}$, each branch first projects them into query, key, and value embeddings:

$$\mathbf{Q}_{ct} = \text{Proj}_{Q_{ct}}(\mathbf{F}_{ct}), \qquad\qquad \mathbf{Q}_{seg} = \text{Proj}_{Q_{seg}}(\mathbf{F}_{seg}), \qquad (1)$$

$$\mathbf{K}_{ct} = \text{Proj}_{K_{ct}}(\mathbf{F}_{ct}), \qquad\qquad \mathbf{K}_{seg} = \text{Proj}_{K_{seg}}(\mathbf{F}_{seg}), \qquad (2)$$

$$\mathbf{V}_{ct} = \text{Proj}_{V_{ct}}(\mathbf{F}_{ct}), \qquad\qquad \mathbf{V}_{seg} = \text{Proj}_{V_{seg}}(\mathbf{F}_{seg}). \qquad (3)$$

Then, a cross-attention operation occurs where each branch queries the other for context:

$$\mathbf{O}_{\text{seg}\leftarrow\text{ct}} = \text{Softmax}\left(\frac{\mathbf{Q}_{\text{seg}}\mathbf{K}_{\text{ct}}^{\top}}{\sqrt{d_k}}\right)\mathbf{V}_{\text{ct}}, \quad \mathbf{O}_{\text{ct}\leftarrow\text{seg}} = \text{Softmax}\left(\frac{\mathbf{Q}_{\text{ct}}\mathbf{K}_{\text{seg}}^{\top}}{\sqrt{d_k}}\right)\mathbf{V}_{\text{seg}}, \qquad (4)$$

where $d_k$ is the key dimension. This information is integrated back via a residual connection and a shared, zero-initialized output projection, $\text{Proj}_{\text{out}}$:

$$\mathbf{F}_{\text{seg}}^{\text{out}} = \mathbf{F}_{\text{seg}} + \text{Proj}_{\text{out}}(\mathbf{O}_{\text{seg}\leftarrow\text{ct}}), \quad \mathbf{F}_{\text{ct}}^{\text{out}} = \mathbf{F}_{\text{ct}} + \text{Proj}_{\text{out}}(\mathbf{O}_{\text{ct}\leftarrow\text{seg}}). \qquad (5)$$

### 3.5 TRAINING STRATEG AND OBJECTIVE

We employ a specialized training strategy to learn a robust semantic alignment.

**Asynchronous Timestep Sampling.**  To prevent the VJA from learning superficial correlations based on identical noise levels, we sample two independent timesteps, $t_{ct}$ and $t_{seg}$, in each training iteration. We add Gaussian noise $\boldsymbol{\epsilon}$ to each latent representation according to its respective timestep:

$$\mathbf{z}_{\text{ct}}^{t_{ct}} = \sqrt{\bar{\alpha}_{t_{ct}}}\mathbf{z}_{\text{ct}} + \sqrt{1 - \bar{\alpha}_{t_{ct}}}\boldsymbol{\epsilon}, \quad \mathbf{z}_{\text{seg}}^{t_{seg}} = \sqrt{\bar{\alpha}_{t_{seg}}}\mathbf{z}_{\text{seg}} + \sqrt{1 - \bar{\alpha}_{t_{seg}}}\boldsymbol{\epsilon}, \qquad (6)$$

where $\bar{\alpha}_t$ is the noise schedule. This forces the VJA to align features from different stages of the diffusion process.

**Joint Training Objective.**  The denoising U-Nets ($\epsilon_\theta^{\text{ct}}, \epsilon_\theta^{\text{seg}}$) are trained to predict the added noise $\boldsymbol{\epsilon}$. The objective is the sum of the L1 losses for both branches:

$$\mathcal{L} = \mathbb{E}_{\mathbf{z},\boldsymbol{\epsilon},t_{ct},t_{seg}}\left[\left\|\boldsymbol{\epsilon} - \epsilon_\theta^{\text{ct}}(\mathbf{z}_{\text{ct}}^{t_{ct}}, t_{ct})\right\|_1 + \left\|\boldsymbol{\epsilon} - \epsilon_\theta^{\text{seg}}(\mathbf{z}_{\text{seg}}^{t_{seg}}, t_{seg})\right\|_1\right]. \qquad (7)$$

Crucially, while the CT U-Net is frozen, its loss term is still required to provide the supervisory signal for training the VJA parameters that modify the feature maps within the CT branch. The resulting gradients update only the trainable parameters: the segmentation branch and all VJA modules.

## 4 EXPERIMENTS

### 4.1 EXPERIMENTAL SETUP

**Datasets and Metrics**  We perform experiments on four 3D CT datasets: our in-house **CVAI** dataset and three public benchmarks, **SegTHOR** Lambert et al. (2019), **MSD-06 Lung**, and **MSD-10 Colon** Antonelli et al. (2022). We specifically selected the MSD tumor datasets as they represent a particularly challenging case due to the high variability in tumor shape, size, and location, making them a prime example of the data scarcity problem our work aims to solve. For all downstream experiments, we created a 30% held-out test set for each dataset to ensure fair comparisons. We evaluate synthesis quality with Fréchet Inception Distance (FID) Heusel et al. (2017), downstream task performance with the Dice Similarity Coefficient (DSC), and structural diversity with Average Pairwise 3D Multi-Scale Structural Similarity (MS-SSIM) on the generated masks Wang et al. (2003); Khader et al. (2022).

**Implementation Details**  The framework was implemented in PyTorch with MONAI library Cardoso et al. (2022). All volumes were resampled to $256^3$ voxels. We train for 250,000 iterations on a single NVIDIA RTX 4090 GPU. The CT LDM branch is frozen during the training. Additional details on hyperparameters and training are in the appendix.

**Baselines**  Our primary baseline for data augmentation is the powerful conditional generator **MAISI-ControlNet**. We use the officially released pre-trained weights for public datasets. For our in-house CVAI data, the baseline was prepared following the official implementation guidelines to handle the custom labels.

## 4.2 EVALUATION OF SYNTHESIS QUALITY

A primary concern when extending a powerful pre-trained generator is the potential degradation of its output quality. This experiment was designed to validate that our architectural additions for joint distribution learning do not compromise the state-of-the-art image fidelity inherited from MAISI.

We assessed synthesis quality on the CVAI dataset by measuring the Fréchet Inception Distance (FID) Heusel et al. (2017) across all three anatomical planes, comparing Med3D-JADE against several baselines. As shown in Table 1, our model not only preserves but slightly improves the image fidelity of the original MAISI model, achieving the best average FID score. This crucial result demonstrates that our method successfully learns the complex joint distribution for mask generation without sacrificing the quality of CT synthesis. Furthermore, the qualitative results in Figure 3 illustrate that our model consistently generates high-resolution, realistic images with finer anatomical details.image branch produce more coherent and realistic anatomy.

Table 1: Comparison of synthesis quality using Fréchet Inception Distance (FID) on the CVAI dataset. Lower is better. Our method preserves and slightly improves upon the competing models, including DDPM Ho et al. (2020), LDM Rombach et al. (2022), HA-GAN Sun et al. (2022), and the current SOTA model-MAISI Guo et al. (2025).

| Method | FID ↓ (Axial) | FID ↓ (Sagittal) | FID ↓ (Coronal) | FID ↓ (Avg.) |
|--------|---------------|------------------|-----------------|--------------|
| DDPM   | 15.03 | 17.23 | 17.77 | 16.68 |
| LDM    | 9.34  | 10.01 | 10.74 | 10.3  |
| HA-GAN | 17.96 | 12.55 | 11.44 | 13.98 |
| MAISI  | 2.01  | 2.42  | 3.09  | 2.51  |
| **Ours** | **1.82** | **2.16** | **2.77** | **2.25** |

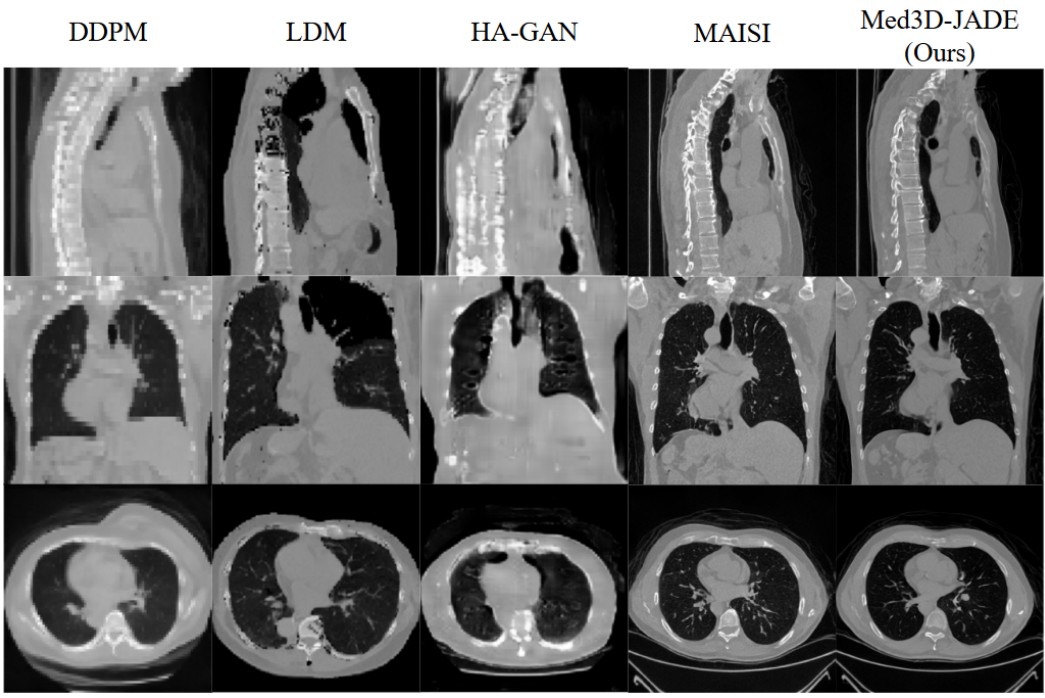

Figure 3: Qualitative comparison of generated CT volumes. Columns compare different generation methods. The rows, from top to bottom, show results for Axial, Coronal, and Sagittal views, respectively. Our Med3D-JADE method demonstrates superior image quality and anatomical fidelity across all views.

## 4.3 EFFICACY FOR DOWNSTREAM AUGMENTATION

The ultimate test of a generative model's utility is its ability to improve downstream clinical tasks. Our core hypothesis is that the **structural diversity** from our joint model offers a fundamental advantage over the **textural variations** from conditional models. We tested this by augmenting the training of three leading segmentation models (nnU-Net Isensee et al. (2021), Swin-UNETR Hatamizadeh et al. (2022), and SegResNet Myronenko (2018)) using a 1:1 augmentation ratio of real to synthetic data. For each segmentation model and dataset, we compared three training schemes: (1) using only real data, (2) augmenting real data with synthetic pairs from the conditional MAISI-ControlNet, and (3) augmenting real data with pairs from our Med3D-JADE.

The results in Table 2 validate our hypothesis. Augmenting with data from Med3D-JADE provides substantial and reliable performance gains across nearly all models and datasets. The trend is particularly evident in the challenging tumor segmentation tasks, underscoring the importance of generating novel structures for pathologies with high variability.

Table 2: Comprehensive downstream segmentation performance (DSC %) across all four datasets. Our method consistently provides the largest performance gain over the real-data-only baseline.

| Dataset | Method | nnUNet | SwinUNETR | SegResNet |
|---------|--------|--------|-----------|-----------|
| CVAI | Real Only | 0.9078 | 0.8785 | 0.8912 |
| | Real + SYN(MAISI) | 0.9233 | 0.8859 | 0.9016 |
| | **Real + SYN(Ours)** | **0.9284** | **0.9088** | **0.9103** |
| SegTHOR | Real Only | 0.9210 | 0.8744 | 0.8719 |
| | Real + SYN(MAISI) | 0.9185 | 0.8737 | 0.8723 |
| | **Real + SYN(Ours)** | **0.9260** | **0.8845** | **0.8837** |
| MSD-06 Lung | Real Only | 0.5265 | 0.5069 | 0.5314 |
| | Real + SYN(MAISI) | 0.5813 | 0.5623 | 0.5444 |
| | **Real + SYN(Ours)** | **0.5822** | **0.5787** | **0.5879** |
| MSD-10 Colon | Real Only | 0.4200 | 0.4134 | 0.4256 |
| | Real + SYN(MAISI) | **0.4501** | 0.4173 | 0.4249 |
| | **Real + SYN(Ours)** | 0.4352 | **0.4192** | **0.4288** |

## 4.4 QUANTITATIVE ANALYSIS OF STRUCTURAL DIVERSITY

To provide direct evidence for our central claim, we quantitatively measure the structural diversity of masks using MS-SSIM (lower is better). The results in Table 3 show that augmenting with Med3D-JADE **consistently produces a more diverse set of masks** than the conditional baseline. Notably, for the CVAI, Lung, and Colon datasets, our augmented set is even more diverse than the original real data. This provides strong quantitative evidence that the structural novelty generated by our model is a key factor in its superior performance as a data augmentation tool.

Table 3: Evaluation of structural diversity on different segmentation mask sets. (Average Pairwise MS-SSIM ↓). A lower score suggests higher diversity. Our method consistently provides a more diverse augmented dataset than the conditional baseline.

| | CVAI | SegTHOR | MSD-06 Lung | MSD-10 Colon |
|---|------|---------|-------------|--------------|
| Real Only | 0.8189 | 0.7671 | 0.9442 | 0.9046 |
| Real + SYN(MAISI) | 0.8198 | 0.7764 | 0.9448 | 0.9051 |
| **Real + SYN(Ours)** | **0.8047** | **0.7705** | **0.9124** | **0.8961** |

## 4.5 QUALITATIVE RESULTS

Figure 4 visually compares real data, pairs from the conditional baseline, and novel pairs from Med3D-JADE. Our generated volumes exhibit high realism, and the masks show precise alignment with the anatomy. More examples showcasing the breadth of generated diversity are in the appendix.

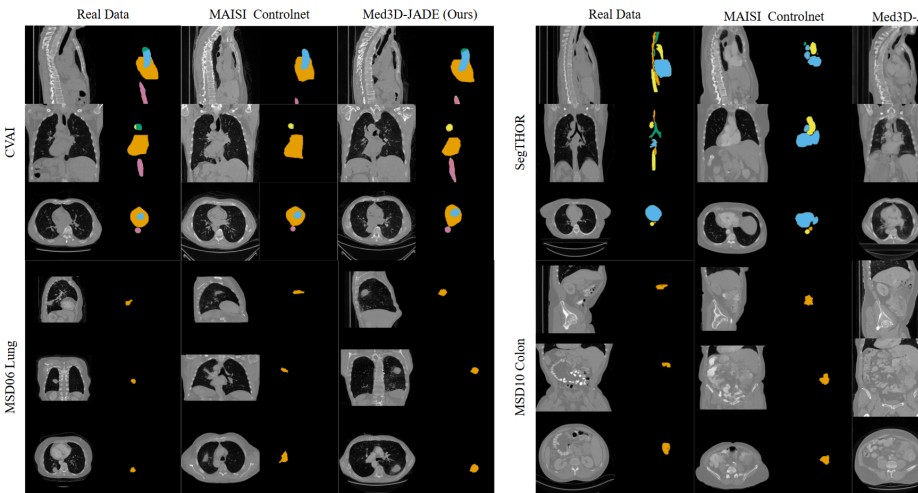

Figure 4: Qualitative comparison of generated pairs. For each dataset, we show a real sample (left), a pair from the conditional MAISI-ControlNet (middle), and a novel pair jointly generated by our Med3D-JADE (right). The figure demonstrates our method's ability to create high-fidelity, anatomically plausible CT volumes with precisely aligned segmentation masks. The full extent of the **structural diversity** unlocked by our joint generation approach, featuring numerous unique samples, is presented in the appendix.

## 4.6 ABLATION STUDIES

### 4.6.1 IMPACT OF SYNTHETIC DATA RATIO

We studied the effect of varying the real-to-synthetic data ratio on a downstream lung tumor segmentation task. The results in Table 4 demonstrate that performance is sensitive to this ratio. This indicates that the quantity of synthetic data is a key hyperparameter that should be carefully tuned to maximize augmentation gains for a given dataset.

Table 4: Data augmentation experiments on the MSD 06 Lung dataset using real and/or synthetic data in terms of DSC on the testing set.

| Real:Synth Ratio | nnUnet |
|---|---|
| 1:0 | 0.5265 |
| 1:0.5 | 0.5602 |
| 1:1 | 0.5822 |
| 1:1.5 | **0.5893** |
| 1:2 | 0.5823 |

Table 5: Impact of VJA on downstream segmentation. Aligned data (with VJA) helps, while misaligned data (w/o VJA) harms performance.

| Training Data | DSC (%) ↑ |
|---|---|
| Real Data Only | 0.5265 |
| **Real+Synth (VJA)** | **0.5822** |
| Real+Synth (no VJA) | 0.4771 |

### 4.6.2 THE CRITICAL ROLE OF VJA

To prove the necessity of our Volumetric Joint Attention (VJA) module, we performed a targeted ablation study. We compared our full Med3D-JADE model against an ablated baseline where the VJA module was removed, causing it to generate misaligned data pairs. We evaluated the utility of synthetic data from both models on a downstream lung tumor segmentation task by augmenting the real training data and comparing against a baseline trained on real data only.

The results, given in Table 5, clearly demonstrate the effectiveness of the proposed VJA module. Adding the synthetic paired data generated by our full model provides a significant performance boost over the baseline. In contrast, adding the misaligned pairs from the ablated model degrades performance, confirming that the network is confused by the incorrect data pairs. Figure 5 depicts a clear qualitative example of this failure case. The model with VJA generates an anatomically plausible lung tumor with a perfectly aligned mask, while the ablated model does not, illustrating the source of the performance drop. This study confirms that the VJA module is essential for producing synthetic data that is not only visually coherent but also beneficial for downstream tasks.

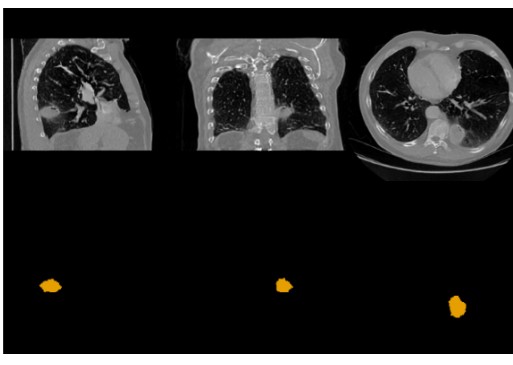 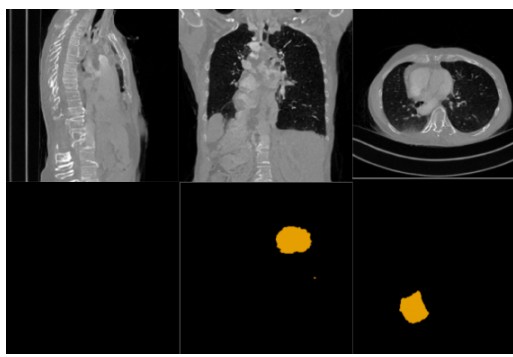

(a) With VJA module        (b) Without VJA module

Figure 5: Qualitative comparison of generated CT volumes and segmentation masks. **(Left) With VJA module:** Demonstrates correct alignment between the generated lung tumor mask and the CT anatomy. **(Right) Without VJA module:** Shows misalignment and inaccuracies in the generated mask when the VJA module is not used.

## 5 CONCLUSION

We presented **Med3D-JADE**, a novel diffusion-based framework that, to the best of our knowledge, is the first to achieve **joint generation of 3D CT volumes and their corresponding segmentation masks**. This approach directly tackles the critical challenge of data scarcity in medical image analysis by learning the joint distribution $p(image, mask)$, which allows for the creation of entirely new, structurally diverse data pairs that are impossible to generate with conditional mask-to-image models.

Our method leverages a powerful pre-trained generator (MAISI) in a dual-branch architecture. By freezing the image branch and training a new parallel segmentation branch, we preserve state-of-the-art image fidelity. Coherence between the two modalities is enforced by our proposed **Volumetric Joint Attention (VJA)** modules. A key innovation is our strategy of mapping discrete mask labels into a continuous space, which allows us to reuse the sophisticated pre-trained Volumetric Compression Network for both branches, enabling faithful, high-resolution synthesis.

We validated our approach by using the generated data to augment training for downstream segmentation tasks. Experiments across the in-house CVAI dataset and several public benchmarks (SegTHOR, MSD Lung, Colon) consistently demonstrated that data augmentation from Med3D-JADE significantly improves the performance of multiple segmentation backbones, including nnU-Net, SwinUNETR, and SegResNet. This demonstrates that the superior **structural diversity** provided by our joint-generation approach translates directly to more robust and generalizable segmentation models. These findings highlight the immense potential of joint generative modeling to alleviate data scarcity and advance the capabilities of medical image segmentation.

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

## A    LLM USAGE

In accordance with ICLR policy, we disclose the use of a large language model to improve the language and clarity of this manuscript. As English is our second language, we used the model to refine sentence structure, check for spelling errors, and ensure grammatical correctness.

## B    ADDITIONAL QUALATATIVE RESULTS OF MED3D-JADE

This section provides additional qualitative results to supplement the main paper, offering further visual evidence of the novel synthetic image-mask pairs generated by Med3D-JADE. The following samples, from the four datasets used in our experiments (Figures 6–9), demonstrate two key capabilities of our model. First, they show that Med3D-JADE consistently generates high-fidelity, anatomically plausible CT volumes with precisely aligned segmentation masks. Second, and more importantly, these visualizations showcase the rich **structural diversity** unlocked by our joint generation paradigm. This directly illustrates the primary benefit of learning the true joint distribution, $p(\text{image, mask})$, which enables the creation of entirely new anatomical structures. This is a fundamental advantage over conditional mask-to-image augmentation methods, which can only produce new textures for the limited set of shapes present in the original training data.

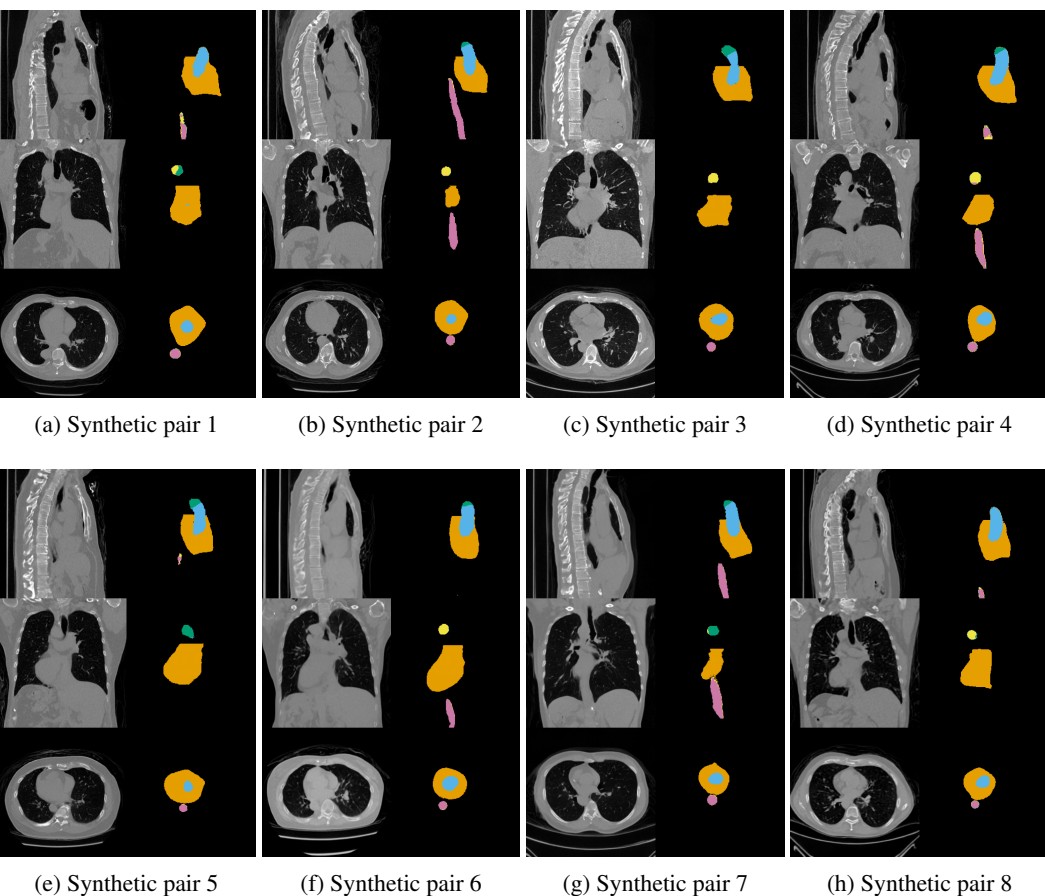

| (a) Synthetic pair 1 | (b) Synthetic pair 2 | (c) Synthetic pair 3 | (d) Synthetic pair 4 |

| (e) Synthetic pair 5 | (f) Synthetic pair 6 | (g) Synthetic pair 7 | (h) Synthetic pair 8 |

Figure 6: Additional qualitative results from Med3D-JADE on **CVAI dataset**, showcasing the generation of diverse and novel anatomical structures with high-fidelity image-mask alignment.

## C    ADDITIONAL DATASET DESCRIPTIONS

**CVAI Dataset:** The **CVAI** dataset is an in-house collection of 150 CT scans annotated by radiology experts. It features six classes (including background) with five clinically significant labels: the **peri-**

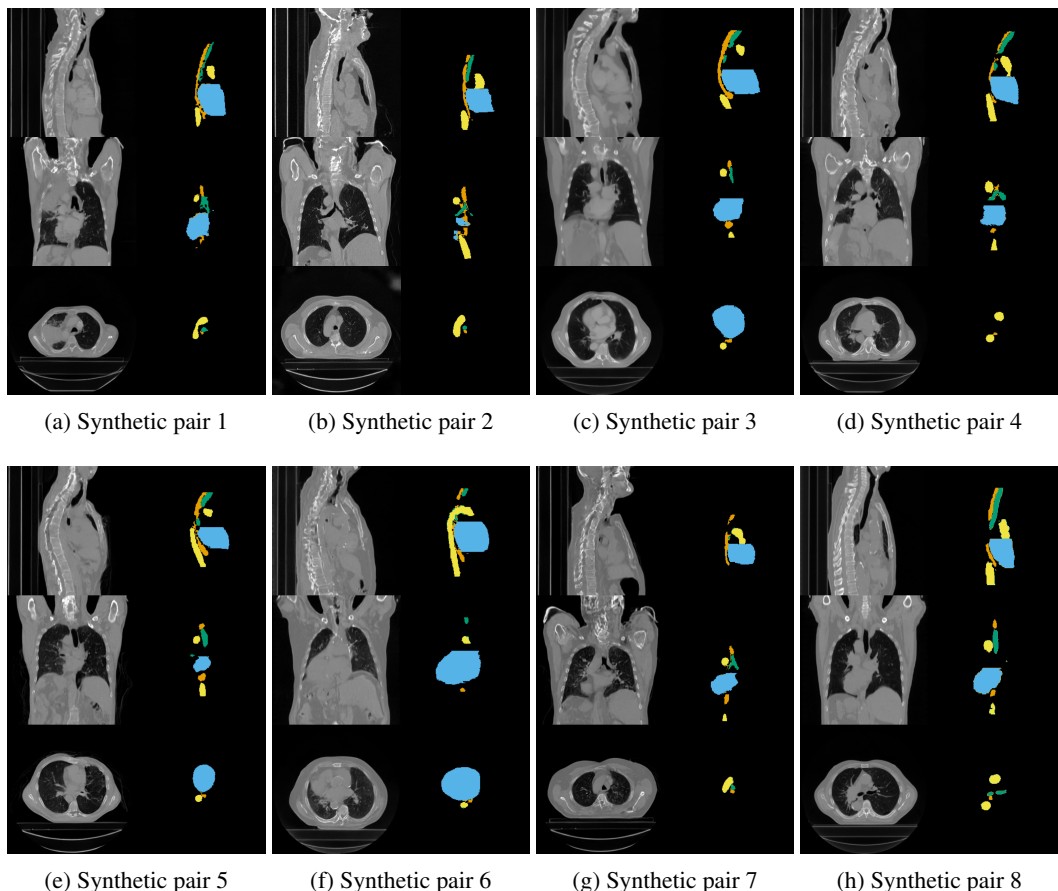

| (a) Synthetic pair 1 | (b) Synthetic pair 2 | (c) Synthetic pair 3 | (d) Synthetic pair 4 |

| (e) Synthetic pair 5 | (f) Synthetic pair 6 | (g) Synthetic pair 7 | (h) Synthetic pair 8 |

Figure 7: Additional qualitative results from Med3D-JADE on **SegTHOR dataset**, showcasing the generation of diverse and novel anatomical structures with high-fidelity image-mask alignment.

**cardium**, the **ascending aorta**, the **aortic arch**, and two separate sections of the **descending aorta**. This fine-grained annotation provides critical diagnostic details often lost in public datasets that use monolithic labels like "heart" or "aorta," making it crucial for diagnosing specific pathologies.

**SegTHOR Lambert et al. (2019):** The **SegTHOR** dataset is a public benchmark for segmenting thoracic organs at risk. It consists of 40 CT scans with five classes (including background), which are the esophagus, heart, trachea, and aorta.

**MSD-06 Lung Antonelli et al. (2022):** To test our model on challenging and heterogeneous tasks, we selected the **MSD-06 Lung** dataset from the Medical Segmentation Decathlon (MSD) challenge. It contains 95 scans and features two classes (tumor and background). This dataset is representative of the data scarcity problem due to the high variability and subtle appearance of lung tumors.

**MSD-10 Colon Antonelli et al. (2022):** The **MSD-10 Colon** dataset, also from the MSD challenge, contains 190 scans. Similar to the lung dataset, it includes two classes (tumor and background) and presents a significant challenge for segmentation models, making it an ideal test case for evaluating the utility of synthetic data augmentation.

Table 6: Experiment Datasets information.

| Dataset | Number of data pairs | Class numbers | Body Region | Modality |
|---|---|---|---|---|
| SegTHOR Lambert et al. (2019) | 40 | 5 | Chest | CT |
| MSD 06 Lung Antonelli et al. (2022) | 95 | 2 | Chest | CT |
| MSD 10 Colon Antonelli et al. (2022) | 190 | 2 | Abdomen | CT |
| CVAI (In-house) | 150 | 6 | Chest | CT |

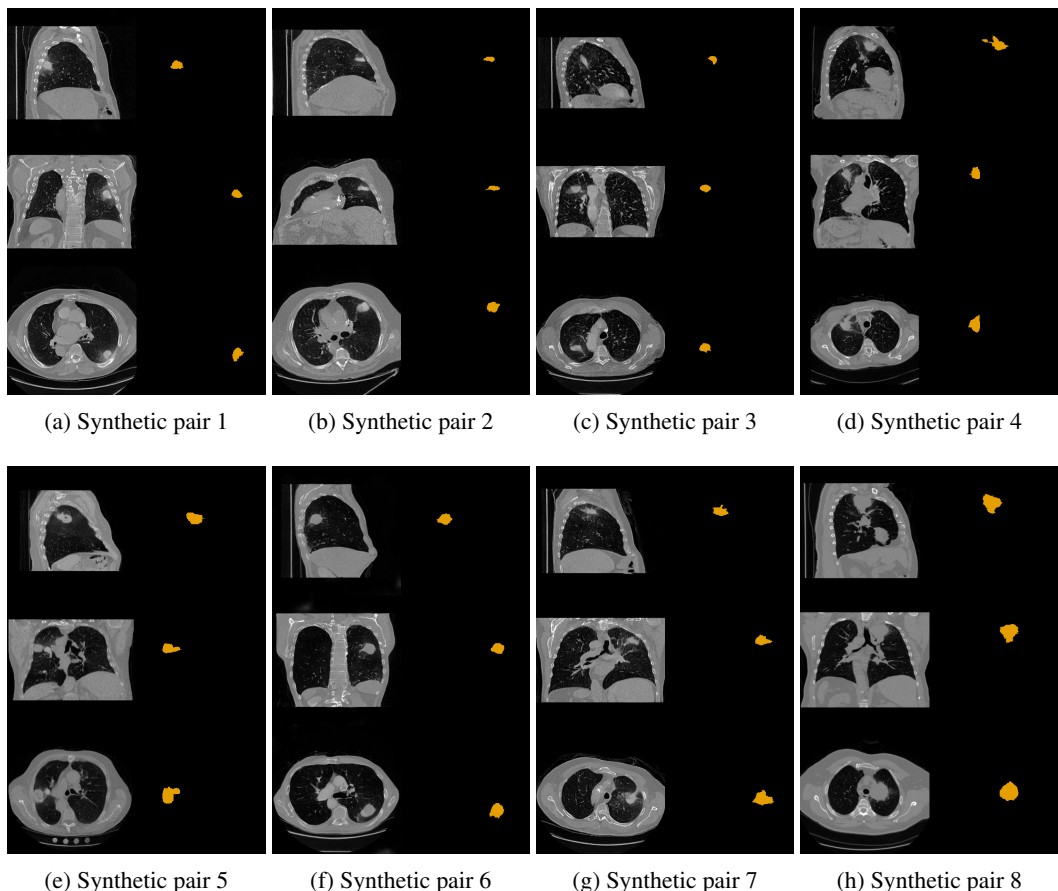

(a) Synthetic pair 1    (b) Synthetic pair 2    (c) Synthetic pair 3    (d) Synthetic pair 4

(e) Synthetic pair 5    (f) Synthetic pair 6    (g) Synthetic pair 7    (h) Synthetic pair 8

Figure 8: Additional qualitative results from Med3D-JADE on **MSD 06 Lung dataset**, showcasing the generation of diverse and novel anatomical structures with high-fidelity image-mask alignment.

## D    ADDITIONAL IMPLEMENTATION DETAILS

### D.1    DETAILED DESCRIPTION OF THE IN-HOUSE CVAI DATASET

Our in-house CVAI dataset consists of 150 CT scans, each meticulously annotated by radiology experts with five clinically-significant labels: the **pericardium**, the **ascending aorta**, the **aortic arch**, and two distinct sections of the **descending aorta**. This expert-level, fine-grained annotation, which required over an hour per scan, addresses a key limitation of public datasets that use coarse "heart" or monolithic "aorta" labels, thereby obscuring critical diagnostic information. The clinical insight gained from this detail is significant. For instance, specifically labeling the pericardium allows for the quantification of pathologies like fluid buildup (effusion) and inflammation (pericarditis). Similarly, the four-part segmentation of the aorta is essential for distinguishing between vascular conditions that demand immediate open-heart surgery and those managed with medication. Consequently, our dataset provides a challenging and clinically-relevant benchmark to evaluate a model's capacity for providing actionable diagnostic insights rather than just general segmentation.

### D.2    IMPLEMENTATION AND FRAMEWORK

The entire framework for Med3D-JADE is implemented using PyTorch. For handling 3D medical data, building model components, and applying transformations, we extensively utilize the MONAI Cardoso et al. (2022) framework.

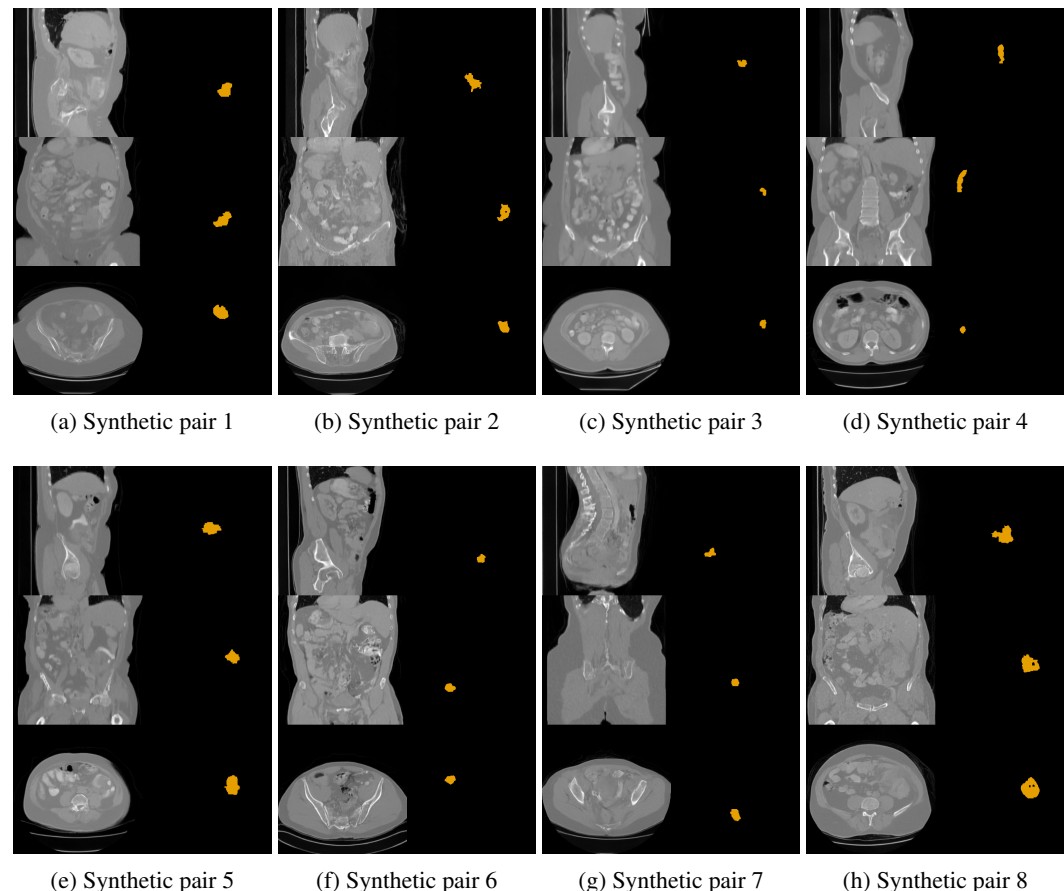

(a) Synthetic pair 1  (b) Synthetic pair 2  (c) Synthetic pair 3  (d) Synthetic pair 4

(e) Synthetic pair 5  (f) Synthetic pair 6  (g) Synthetic pair 7  (h) Synthetic pair 8

Figure 9: Additional qualitative results from Med3D-JADE on **MSD 10 Colon dataset**, showcasing the generation of diverse and novel anatomical structures with high-fidelity image-mask alignment.

### D.3 DATA PREPROCESSING

All volumetric data from the different datasets are preprocessed uniformly to ensure consistency. Each volume is resampled to a fixed spatial resolution of $256 \times 256 \times 256$ voxels. For intensity normalization, we follow the protocol established by MAISI. Specifically, the Hounsfield Unit (HU) values of each CT scan are first clipped to the range $[-1000, 1000]$ and subsequently normalized to a floating-point range of $[0, 1]$.

### D.4 MODEL ARCHITECTURE

The architectural backbones of our 3D Latent Diffusion Models (LDMs) for both the CT and segmentation branches, as well as the underlying Variational Autoencoder (VAE) of the Volume Compression Network (VCN), are based on the official MAISI implementation.

### D.5 TRAINING DETAILS

Med3D-JADE was trained using automatic mixed-precision ('fp16') to accelerate computation and reduce memory usage. We used a single NVIDIA RTX 4090 GPU with a batch size of 2. For each dataset, the model was trained for a total of 250,000 iterations. Key hyperparameters and settings are as follows:

- **Optimizer:** We used the AdamW optimizer.
- **Learning Rate:** The initial learning rate was set to $1 \times 10^{-4}$.

- **Scheduler:** A Cosine Annealing learning rate scheduler was employed, with a linear warm-up phase of 5,000 steps.
- **Loss Function:** The model was optimized using the L1 loss, as described in the main paper.
- **Conditional Embeddings:** The MAISI 3D LDM architecture requires additional conditional embeddings. During training, we set the `top_region_index` to `[0,1,0,0]` and the `bottom_region_index` to `[0,0,1,0]` for all datasets. The voxel spacing condition was dynamically set to match the spacing of each training sample after it was resized.

### D.6 INFERENCE DETAILS

During inference, new image-mask pairs were generated using a DDPM sampler with 1000 steps. The region condition embeddings (`top_region_index` and `bottom_region_index`) remained the same as those used during training. The voxel spacing condition, however, was set to a fixed value depending on the target dataset to ensure consistent generation.

- For the **CVAI** and **SegTHOR** datasets, we used a spacing of $[1.5, 1.5, 1.5]$.
- For the **MSD-06 Lung** and **MSD-10 Colon** tumor datasets, we used a spacing of $[1.5, 1.5, 2.0]$.

## E  ADDITIONAL IMAGE QUALITY EVALUATION

### E.1  FID EVALUATION ON OTHER DATASETS

We also conduct an FID comparison between MAISI and our method on the other three public datasets.

| Dataset | Method | FID ↓ (Axial) | FID ↓ (Sagittal) | FID ↓ (Coronal) | FID ↓ (Avg.) |
|---------|--------|---------------|------------------|-----------------|--------------|
| Colon   | MAISI  | 8.909         | 7.182            | 13.286          | 9.792        |
|         | **Ours** | **1.904**   | **2.447**        | **3.85**        | **2.734**    |
| SegTHOR | MAISI  | 10.433        | 8.741            | 14.404          | 11.193       |
|         | **Ours** | **1.559**   | **1.604**        | **3.02**        | **2.061**    |
| Lung    | MAISI  | 2.169         | 3.527            | 9.253           | 4.983        |
|         | **Ours** | **1.323**   | **1.363**        | **8.013**       | **3.566**    |

Table 7: Comparison of synthesis quality using Fréchet Inception Distance (FID) across multiple datasets. Lower is better. Our method improves upon the current SOTA model MAISI.

