# OpenReview forum: "Med3D-JADE: 3D Joint Attentive Diffusion Engine for Volumetric Medical CT and Mask Co-generation"
_ICLR.cc/2026/Conference — ICLR 2026 Conference Withdrawn Submission_

### Official Review · Reviewer_ryj6 · 2025-10-16

**Soundness:** 3
**Presentation:** 2
**Contribution:** 2
**Rating:** 2
**Confidence:** 4

**Summary:**

This paper builds upon the MAISI 3D medical diffusion model and extends it to achieve joint generation of CT volumes and their corresponding segmentation labels. The authors introduce a dual-branch framework that consists of a frozen CT generation branch inherited from MAISI and a newly trained segmentation branch. The two branches interact through a Volumetric Joint Attention (VJA) mechanism to ensure anatomical coherence between modalities. Additionally, the method employs asynchronous timestep sampling and a joint training objective to encourage semantic alignment while preventing trivial noise-level correlations.

**Strengths:**

**Extension to label generation.** The work extends 3D diffusion modeling from CT image synthesis to joint label generation, addressing a practical and under-explored problem in medical data synthesis.

**Lightweight and modular design.** The proposed Volumetric Joint Attention (VJA) module is compact and easily integrated into pretrained diffusion models without retraining the image branch.

**Practical data augmentation benefit.** The jointly generated CT–mask pairs improve segmentation performance across multiple datasets and backbones, demonstrating practical value.

**Weaknesses:**

**Efficiency and Memory Reporting**
The dual-branch diffusion framework should introduce additional memory and computational cost. However, the paper does not report GPU memory usage, training or inference time, or efficiency comparisons. Quantitative efficiency analysis is necessary for practical assessment.

**Baseline Construction and Fairness**
While the paper claims to be the first to achieve single-stage joint generation of CT and segmentation volumes, the baselines are limited to conditional generation models (e.g., MAISI-ControlNet) and only evaluate CT image quality (FID). The comparison omits quantitative assessment of the generated labels and excludes existing paired-generation works, making the evaluation insufficient to fully validate the “joint generation” claim.

**Frozen CT Branch and Training Scope**
During training, the CT generation branch is completely frozen and inherited from MAISI, while only the segmentation branch and VJA modules are updated. As a result, most of the image quality is preserved from MAISI rather than learned through the proposed joint framework. The slightly better FID could stem from minor effects of VJA modules or statistical variation, rather than genuine improvement in generative capability.

**Questions:**

1. In Table 1, only FID is reported. Could the authors include additional metrics (e.g., PSNR, SSIM, or clinical evaluation) to better demonstrate performance?

2. The paper mainly compares with MAISI. Although MAISI is a strong baseline, could the authors compare with more models (e.g., MedGen3D, MedSegFactory) to strengthen the empirical validation?

3. Given that the CT branch is frozen, how does the model achieve a slightly better FID than MAISI? Does the improvement result from the VJA modules or metric variation?

4. The proposed dual-branch structure likely increases memory and computation. Could the authors report training/inference time and GPU memory usage?

---

### Official Review · Reviewer_sDj9 · 2025-10-26

**Soundness:** 3
**Presentation:** 3
**Contribution:** 2
**Rating:** 4
**Confidence:** 4

**Summary:**

This paper introduces a novel diffusion-based framework that simultaneously generates realistic 3D CT scans and their corresponding segmentation masks, addressing the challenge of limited annotated medical data. Unlike traditional conditional methods that generate images from fixed masks, Med3D-JADE learns the true joint distribution of image–mask pairs, enabling the creation of new anatomical and pathological structures. The model builds on the MAISI foundation model using a dual-branch latent diffusion architecture—one frozen for CT generation and one trained for segmentation—linked by a Volumetric Joint Attention (VJA) module to ensure spatial and semantic alignment. Experiments on multiple datasets, including SegTHOR and MSD tumor sets, show that Med3D-JADE improves image fidelity, enhances mask diversity, and significantly boosts downstream segmentation model performance, making it a powerful tool for medical data augmentation and synthesis.

**Strengths:**

The method provides a well-structured integration of generation and segmentation components, which helps clarify the overall workflow.
The authors conduct multiple experiments across different datasets, demonstrating the method’s potential applicability in various medical imaging scenarios.

**Weaknesses:**

I am concerned about the generalization capability of this method, as the authors utilize a pre-trained generation structure. This frozen structure may not fit all types of medical data.

The method appears to add a segmentation head to obtain mask results. Why, then, is the title “co-generation”?

The attention-based design lacks novelty in 2025.

The datasets are already outdated.

**Questions:**

N/A

---

### Official Review · Reviewer_MbQG · 2025-10-27

**Soundness:** 3
**Presentation:** 3
**Contribution:** 2
**Rating:** 2
**Confidence:** 4

**Summary:**

The paper presents a technically plausible method for adapting a foundation model for joint image-mask synthesis, centered on the Volumetric Joint Attention (VJA) module. However, the central claims of the paper are not convincingly supported by the provided evidence. The main validation—downstream segmentation improvement—shows highly inconsistent and often marginal gains, with the conditional baseline (MAISI) even *outperforming* the proposed method on a public tumor dataset. Furthermore, the core claim of generating clinically meaningful "structural diversity" is unsubstantiated; it is evaluated using a weak proxy metric (MS-SSIM) without any assessment of anatomical or pathological plausibility, which is a critical omission for a medical imaging application. The work's novelty is also limited, presenting more as an incremental adapter for MAISI than a generalizable new framework.

**Strengths:**

Methodological Component: The ablation study in Table 5 and Figure 5 convincingly demonstrates that the proposed VJA module is essential for enforcing image-mask alignment . Augmenting with data from the VJA-ablated model severely degrades performance, confirming the module's necessity within this architecture.

Parameter-Efficient Design: The strategy of freezing the high-fidelity CT branch and reusing the pre-trained VCN for both modalities is a practical and parameter-efficient approach to leveraging large foundation models .

Downstream Task Validation: The authors test the synthetic data's utility by augmenting three different, state-of-the-art segmentation backbones (nnU-Net, SwinUNETR, SegResNet), demonstrating some performance gains across all four tested datasets.

**Weaknesses:**

- The paper's primary evidence for its utility is weak and contradictory. On the public SegTHOR dataset, the gains are marginal for the best model (nnU-Net: 0.9210 to 0.9260). On the public MSD-10 Colon tumor dataset, the conditional baseline (Real+SYN(MAISI)) *outperforms* the proposed joint model (Real+SYN(Ours)) on the nnU-Net (0.4501 vs 0.4352). The most significant gains are achieved on the in-house, non-public CVAI dataset. This reliance on unverifiable data for its strongest results is a major weakness.
- Unsubstantiated Claim of "Structural Diversity": The paper's central hypothesis is that it generates "novel structural diversity". This claim is supported *only* by a lower average pairwise MS-SSIM on generated masks. This metric is a poor proxy for clinical utility. A lower MS-SSIM score does not differentiate between *plausible* novel anatomy/pathology and *implausible* artifacts, noise, or unrealistic shapes. There is no evidence that the generated diversity is clinically meaningful.
- Complete Lack of Clinical Plausibility Assessment: For a paper claiming to generate novel anatomical and pathological structures (especially tumors, as in MSD-06 and MSD-10), the absence of any qualitative or quantitative assessment by medical experts is a severe omission. FID (Table 1, 7) measures perceptual fidelity of the *image*, not the *anatomical correctness* of the image-mask pair. The qualitative examples (e.g., Figure 8, 9) are insufficient to judge if these are realistic new tumor presentations or statistical artifacts.

**Questions:**

See Weakness.

**Details Of Ethics Concerns:**

No Ethics Concerns.

---

### Official Review · Reviewer_SDub · 2025-10-31

**Soundness:** 2
**Presentation:** 3
**Contribution:** 2
**Rating:** 4
**Confidence:** 4

**Summary:**

The paper introduces Med3D-JADE, which is aiming to generate 3D paired image-mask pairs. The paper performs evaluation on both generation and segmentation, demonstrating the effectiveness of using Med3D-JADE as an augmentation tool.

**Strengths:**

1. The paper is well-motivated by one of the key challenges in medical image segmentation.
2. The paper presents comprehensive quantitative and qualitative results.
3. The paper evaluates on multiple CT datasets.

**Weaknesses:**

1. The paper misses one of the main related work MedSegFactory [1], which although is developed in 2D, the key designs are very similar (e.g., by comparing JCA and VJA). Also, the network heavily depends on MAISI, therefore, the novelty of the paper remains a concern.
2. What would be the performance if the model is trained from scratch (not initialized with MAISI)
3. How the model is compared with MAISI in table 2, where MAISI is trained on unlabeled CTs. More details are needed.

[1] Mao, Jiawei, et al. "Medsegfactory: Text-guided generation of medical image-mask pairs." arXiv preprint arXiv:2504.06897 (2025).

**Questions:**

N/A

**Details Of Ethics Concerns:**

Will the code be released?

---

### Note · Authors · 2026-03-11

I have read and agree with the venue's withdrawal policy on behalf of myself and my co-authors.

---

### Meta-Review · Area_Chair_d8A1 · 2025-12-23

**Summary:**

This paper presents a 3D joint attentive diffusion engine for CT and mask co-generation. The reviewers raised some major concerns, including limited novelty ( Reviewer SDub, Reviewer sDj9), marginal improvement (Reviewer MbQG ), clinical significance (Reviewer MbQG), insufficient experiments (Reviewer ryj6).

**Reviewer Concerns:**

The major concerns include limited novelty ( Reviewer SDub, Reviewer sDj9), marginal improvement (Reviewer MbQG ), clinical significance (Reviewer MbQG), insufficient experiments (Reviewer ryj6).  There were no rebuttals provided by the authors.

**Reviewer Scores:**

I do not think the reviewers would change their scores as there are no rebuttals provided.

---

### Decision · Program_Chairs · 2026-01-26

Reject